# Beyond the healthcare system: The societal and contextual factors impacting parents' participation in decision-making for neonates with life-threatening conditions

Fatemeh Oskouie[1,2], Sedigheh Khanjari[1,2], Marjan Banazadeh[2,3,4¤] *

1 Nursing and Midwifery Care Research Center, Health Management Research Institute, Iran University of Medical Sciences, Tehran, Iran, 2 School of Nursing and Midwifery, Iran University of Medical Sciences, Tehran, Iran, 3 Department of Nursing, School of Nursing, Alborz University of Medical Sciences, Karaj, Iran, 4 Nursing and Midwifery Care Research Center, Iran University of Medical Sciences, Tehran, Iran

¤ Current address: Alborz University of Medical Sciences, Karaj, Iran
* banazadehmarjan@gmail.com, banazadeh54@yahoo.com

## Abstract

### Background

Parents of neonates with life-threatening conditions and professionals, bear the burden of making complex decisions. Parents may not be fully involved in decision-making, and there is a paucity of evidence regarding the influence of social context on parents' participation. We aimed to explore factors that extended beyond the healthcare system and impacted parents' participation in decision-making for neonates with life-threatening conditions.

### Materials and methods

This qualitative research was carried out in 2019 in four level-III Iranian NICUs, (neonatal intensive care units) where twenty-three face-to-face semi-structured interviews were conducted, transcribed, and analyzed using a conventional content analysis technique. Interviews were condensed into meaningful units during the coding phase, resulting in 206 open codes. These codes were then categorized into eleven categories based on commonalities and distinctions. This iterative process continued until 4 main subcategories were established.

### Results

The main categories and sub-categories were "**unmodified regulations according to the neonatology advances**" (lack of regulations to modify ineffective treatments, lack of a legally documented do not resuscitate order, lack of a defined regional neonatal viability threshold, and lack of maternal guardianship of child medical care), "**deficiencies of the health insurance system**" (covering the cost of ineffective treatments and lack of insurance covering for palliative care services), "**treatment-oriented culture in society**" (expecting a miracle for medical science, difficult acceptance of neonatal death and difficult

**Data Availability Statement:** All relevant data are within the manuscript.

**Funding:** The author(s) received no specific funding for this work.

**Competing interests:** The authors have no financial relationships relevant to this article to disclose." "Potential Conflicts of Interest: The authors have no conflicts of interest relevant to this article to disclose." "Funding: none.

**Abbreviations:** DM, decision making; DNR, do-not-resuscitate; EOL, end of life; FCC, family-centered care; LTC, life-threatening conditions; NICU, neonatal intensive care unit; NPC, neonatal palliative care; SDM, shared decision making.

acceptance of home death), and "**physician-oriented culture in society**" (excessive respect for physicians' decision-making eligibility and social position of physicians).

## Conclusion

The findings revealed concepts surrounding parents' participation in decision-making for life-threatening conditions neonates are influenced by social, legal, cultural, and financial aspects. To bridge the gap between healthcare professionals' attitudes and cultural and religious beliefs, fatwas, and laws, a collaborative approach is necessary. To address the complex challenges of decision-making for these neonates, involving stakeholders like clinicians, legal experts, Islamic scholars, sociologists, jurists, judges, and medical ethicists is crucial for modifying laws to align with neonatology advancements.

## Introduction

Parents of neonates with life-threatening conditions (LTC), alongside professionals, bear the weighty responsibility of making crucial decisions [1]. The process of making decisions in these situations can be complex and involve medical, social, and ethical considerations. Parents should have the right to be involved in decision-making (DM) regarding their child's healthcare [2]. Decisions concerning children are steered by principles of collaborative DM, whereby the parent and clinician are regarded as being on a status that seems to be of equal value [3]. The use of shared decision-making in pediatrics specifically for children with medical complexity is not well-understood [4] and parents' involvement in DM is not sufficiently implemented in clinical settings [5]. In Iran, the importance of parents' involvement in making healthcare decisions for their children is not emphasized, and families face challenges when dealing with their neonates who have LTC [6].

The involvement of parents in DM for neonates with LTC conditions [7] is related to factors including ethical principles, morality, values, beliefs, standards, legal considerations, and personal and professional experience [8]. Research in the area of neonates with LTC has identified a range of factors that can be classified into several dimensions, encompassing aspects related to parents, neonates, organizations, and healthcare professionals. Among the dimensions associated with parents, key factors include parental competencies, self-efficacy, beliefs, and living conditions [9, 10]. Neonatal-related dimension encompasses structural malformations, gestational age, and chromosomal abnormalities [11]. Organizational-related dimension encompasses power imbalances, deficiencies in ethics committees and hospital regulations, limited resources, and limited choices for ongoing hospitalization [12]. The dimension related to healthcare professionals encompasses various factors, including risk aversion, information deficiencies, and clashes of attitudes [13].

Regarding DM in LTC, the perception of what is morally acceptable or unacceptable in the healthcare environment can vary depending on the socio-cultural context [14] which can differ greatly among various societies [15]. In addition to the organizational aspects of the health system and the individuals involved in the DM process, it is important to address the factors related to the societal context. On the other hand, legal representations are strongly connected to the social context in which they exist, and as a result, they are both shaped by and have an impact on that context [16].

There is a lack of research that explores the societal context that can impact parents' involvement in DM in neonatal healthcare settings. It is important to gain a more comprehensive understanding of the phenomena. Qualitative research offers a cultural and contextual explanation and understanding of social phenomena that cannot be obtained through quantitative research methods. It is suitable for examining the process of healthcare DM, the context in which it takes place, and the perspectives of those involved [17]. This qualitative study aimed to explore the factors that extended beyond the healthcare system and impacted parents' participation in DM for neonates with LTC.

## Materials and methods

### Study design

This multicenter study applied a qualitative content analysis approach to explore the factors that extended beyond the healthcare system and impacted parents' participation in DM for neonates with LTC in 2019, in Iran.

### Participants and settings

The process of DM is a social and interactive phenomenon that involves multiple individuals. The study included a total of twenty-three participants who were either directly or indirectly engaged in DM for neonates with LTC. The study involved a cohort comprising 10 parents (5 mothers and 5 fathers) of hospitalized neonates, 4 nursing professionals, and 6 neonatologists employed in the NICUs. Among the neonatologists, one individual held a position as an official within the Neonatal Health Office of the Ministry of Health and Medical Education (MOHME). Additionally, the participants consisted of an insurance agent, a forensic physician, and, and a Jurisprudence and Principles of Islamic Law (JPIL) expert. All participants were Iranian and spoke Farsi. They were selected based on their willingness to participate in the study and share their experiences. The professionals included in the study had at least 1 year of clinical experience in neonatal healthcare settings. The research was carried out in 4 level-III NICUs of governmental and educational hospitals affiliated with Iran, and Tehran Medical Sciences Universities. In these centers newborns who were either ≤28 days old or weighed ≤2.5 kg were admitted. All the centers offered care to newborns who were at least twenty-four weeks gestational age and in 2 centers newborns who required surgical procedures were hospitalized. None of the NICUs had the capability of providing care in heart surgeries.

In this study, purposeful sampling [18] was used to select parents of neonates with LTC such as prematurity, complex congenital anomalies, and severe asphyxia [17]. The researchers conducted in-depth, semi-structured, face-to-face interviews. The participants agreed on the location, date, and time that were chosen. The interviews took place between September 2018 and October 2019 and lasted 43 to 102 minutes. The participants gave their permission for the interviews to be recorded. The team of researchers talked about whether the questions were suitable. The interviews started by asking a general question tailored to each participant and then followed up based on their responses (Table 1). Furthermore, probing questions were asked to gain a deeper understanding of the phenomenon. To provide an accurate description and interpretation of the participant's responses, field notes were recorded during the interviews. The researchers conducted a total of twenty-four interviews, continuing until they reached data saturation, and determined that additional interviews would not yield any new data.

**Table 1. The interview guides.**

| Interview guide of parents |
|---|
| 1. Please talk about your baby's condition. |
| 2. How did you participate in decisions being made for your neonate? |
| 3. How would you like to be involved in these decisions? |
| 4. What barriers prevent you from participating in DM? |
| 5. What are the facilitators of your participation in DM? |
| **Interview guide of health care Professionals** |
| 1. How do you involve parents in DM? |
| 2. What barriers prevent parents from participating in DM? |
| 3. What are the facilitators of parents' participation in DM? |
| **Interview guide of individuals indirectly involved in DM** |
| 1. What do you think about parents' participation in DM for neonates with life-threatening conditions?? |
| 2. How do health system policies prevent parents' participation in DM? |
| 3. How do existing laws and rules prevent parents' participation in DM |

## Reliability and validity

To establish trustworthiness, the criteria proposed by Lincoln and Guba (1985) [17] such as credibility, confirmability, dependability, and transferability were used. The credibility of the study was improved by closely observing and interacting with the participants over 9 months [19]. For member checking a summary of the codes and themes that emerged from the data was shared with the participants for data verification. Additionally, data and method triangulation were employed to enhance the credibility of the results [17]. Field journaling was used as a means to implement reflexivity strategies. Reflexivity enhanced confirmability to avoid researchers' professional experiences influencing their interpretation of events [19]. Four unbiased faculty members assessed the procedure to verify the dependability of the study [17]. Documentation of the decisions made during data generation and analysis ensured the transferability of the findings and also served as an inquiry audit for other researchers who may want to use the same data [19].

## Ethical consideration and consent to participate

The study was approved by the ethics review committee at Iran University of Medical Sciences (IR.IUMS.REC.1397.388). The objectives of the study were initially explained verbally and in writing, and informed consent was obtained. To protect the identities of the participants, pseudonyms were used. Participants were guaranteed the confidentiality of their data, the right to refuse or withdraw at any point, and were informed that their participation was voluntary and anonymous.

## Data analysis

A qualitative content analysis with a conventional approach was used to analyze data. Data collection and analysis were conducted simultaneously. The researchers transcribed the recorded interviews word for word. Next, the process of inductive content analysis was carried out in three stages: preparation, organization, and reporting [20]. In the initial phase of preparation, each transcribed interview or field note was treated as an individual data unit, and the researchers engaged in thorough readings of these materials on several occasions to gain a comprehensive understanding of the data. The organizing phase consisted of 3 steps: open

coding, category creation, and abstraction. In the phase of open coding, labels were created utilizing the participants' own language to develop succinct meaning units. Then initial codes were grouped into subcategories based on similarities and differences. The number of categories was reduced by combining similar ones into more general categories at a higher level of abstraction resulting in a comprehensive abstract description. This was achieved by moving back and forth between the categories and subcategories. During the reporting stage, the findings of the study were prepared for publishing. The research team reached an agreement on the coding and categorization process. The analysis was managed in MAXQDA software [10].

## Results

### Demographic characteristics of participants

The ages of the parents varied between 27 and 49 years. Among them, one parent possessed a basic education, 6 held diplomas, and one had completed a bachelor's degree. Furthermore, 2 individuals received education beyond the level of a bachelor's degree. Of the mothers, 4 were homemakers while one was in employment. Regarding the fathers, three were self-employed, one was a General Physician, and one worked as a teacher. Three parents had faced infertility issues, with 5 undergoing various treatments. Five parents had only the hospitalized neonate as their child, while 3 parents had one additional child each, and 2 parents had 2 additional children each. The ages of the infants at the time of the interviews ranged between 8–44, with gestational ages between twenty-five and thirty-nine weeks. Among the newborns, there were 5 boys, four girls, and one with ambiguous genitalia. The newborns' hospitalization length during the interviews varied between 6–93 days.

Other participants' ages ranged between27-59 years old. There were 8 women and 5 men, 12 married and one single. One had no children, f5 had one child, and 7 had 2 children. Their work experience ranged between 5–29 years, with NICU experience ranging between 5–25 years.

### Main results

Data analysis resulted in establishing 206 final codes, which were organized into 11 categories, 4 subcategories, and a singular overarching theme (Table 2).

### Beyond the healthcare system

Results of interviews with individuals who were directly and indirectly involved in DM for neonates with LTC revealed that certain factors influenced parents' engagement in DM. These factors were not related to the structure of healthcare organizations or providers but rather were influenced by the legal and cultural context of the society. This overarching theme consisted of four subcategories, each arising from various categories.

**Unmodified regulations according to the neonatology advances.** Some legal limitations prevented HPs from involving parents in the DM process due to the inconsistency between laws and advancements in newborn care. This subcategory emerged from the 4 following categories.

*Lack of regulations to modify ineffective treatments*. The participants believed that although they were using current technologies to keep critically ill infants alive, they faced difficulties in transitioning from invasive to non-invasive treatments due to a lack of clear regulations on when to stop ineffective treatments. As a result, they were unable to effectively communicate honest information to parents about their neonate's condition promptly, hindering their involvement in the decision-making process. Participants stated: "*We have an extremely ill*

**Table 2. The societal and contextual factors affecting parents' participation in DM.**

| Example of participants' quotations | Categories | Sub-categories | Main theme |
|---|---|---|---|
| *"We are required to do all treatments. In cases where parents refuse ineffective invasive treatments, the neonatologist defers to the forensic physician to make the decision. This is to prevent any legal complications. If the forensic physician orders to proceed with the procedure, the parents have no say in the matter" (nurse-18).* | Lack of regulations to modify ineffective treatments | Unmodified regulations according to the neonatology advances | **Beyond the healthcare system** |
| *"Issuing the non-resuscitation order is against the law. Occasionally, CPR is ineffective and we do not perform it, but we are still required to fill out the CPR form. We pretended we did CPR and Many deaths are attributed to unsuccessful CPR on record, but in reality, they occur due to the decision to cease futile CPR attempts" (neonatologist-23).* | Lack of a legally documented do not resuscitate order | | |
| *"According to the texts, neonates over 24 weeks gestational age are viable, and caring for this neonate is significantly challenging. While some NICUs may have the resources to provide high-quality care to them, remote areas may lack these capabilities. sometimes a 25-week-old baby may be very ill and parents may not afford NICU costs. Legal obligations dictate that the child is viable and every effort must be made to sustain the child. However, if legally protected, we could inform the family if the child's condition does not improve and treatment proves ineffective* (neonatologist-16). | Lack of a defined regional neonatal viability threshold | | |
| *"Mothers are not given the authority to provide consent, while fathers have this right. Despite the physical and emotional challenges mothers endure during pregnancy, their opinions are not sought or valued. I like both my husband and I to be consulted. Is it only the father's opinion that matters? What about mine? Am I being disregarded"* (mother-4). | Lack of maternal guardianship of child medical care | | |
| *Insurance services influence decisions, in the intubated neonate, the insurance covers all expenses. In this scenario, even if it is not effective, doctors may not feel as pressured to make decisions for the patient. The patient remains on life support until passing away, parents are not financially burdened to make a decision. This may make it easier for them to feel that they have done the best for their child" (insurance company agent-19).* | Covering the cost of ineffective treatments | Deficiencies of the health insurance system | |
| *"It could be more cost-effective for insurance companies to cover palliative care. Insurance costs are determined by the services and treatments prescribed by doctors. There may be a requirement for a more thorough evaluation of the expenses associated with palliative care, rather than continuing with ineffective and intrusive treatments. When palliative care is done legally, officially, and documented, then its services are covered by insurance"* (Insurance company agent-19). | Lack of insurance covering palliative care services | | |
| *"Currently, if I didn't have a seriously ill neonate with a complex illness and wasn't personally affected by this issue, my perspective might be different. Like many others, I might have said no, it's a shame, all efforts to treat the baby should persist because now science has progressed. Well, now I find myself in a difficult situation, I can comprehend more deeply that continuing the treatment would be futile. Without firsthand experience of these difficulties, one cannot make a decision. I never anticipated being caught up in this dilemma" (father-6)* | Expecting a miracle for medical science | Treatment-oriented culture in society | |
| *"During my wife's pregnancy, we shopped daily, purchasing clothes, toys, and a crib for our baby. We also prepared his room, and our family eagerly awaited his arrival. How can we say that our child will not survive" (father-6)* | Difficult acceptance of neonatal death | | |
| *"in our culture, families are reluctant to bring the child home to pass away. They would rather have the child remain in the hospital and pass away there"* (nurse-12). | Difficult acceptance of home death | | |
| *Parents often have difficulty talking to doctors and may not express their disagreement. However, they are more likely to openly disagree with nurses, providing multiple reasons and becoming angry. This behavior may be influenced by the societal emphasis on the importance of doctors"* (nurse-18). | Excessive respect for physicians' decision-making eligibility | Physician-oriented culture in society | |
| *"The influence of a country's culture also plays a role. In my opinion, societal awareness has not advanced sufficiently to facilitate parental involvement in decision-making processes. in DM. It has become so common in our culture that we doctors do everything for the benefit of the patient. Most of the decisions are made by the doctors"* (neonatologist-11). | Social position of physicians | | |

*neonate with severe asphyxia and frequent seizure episodes who is currently under the ventilator. However, only the heart beats, and even the pupils do not respond. The parents desire to stop the ventilator, but it is against the law to do so, even with parents' consent"* (neonatologist-16).

*"There is a gap in the legal system that needs to be addressed by reviewing and revising laws. The hospital's ethics committees should identify and report the challenges presented by these cases to lawmakers"*

*(forensic physician-20)*

*Lack of a legally documented do not resuscitate order.* The lack of a legally documented do-not-resuscitate (DNR) order impacted the involvement of parents in DM. Participants believed that a documented order stating not to resuscitate in cases where it would be ineffective was not considered valid. Consequently, the fear of potential legal action led to verbal discontinuation of resuscitation efforts and prevented HPs from including parents in the DM process. Participants stated: *"Even if the parents do not agree, the medical team is mandated to try until the last moment because it is obligatory to preserve life. Many times they do CPR and the child will suffer until the child dies and there is nothing they can do"* (Associate Professor of JPIL-21).

*In certain situations, the patient may not be resuscitated. but this decision is kept confidential. A doctor may verbally order not to resuscitate the patient without a written order due to concerns about complaints from the family. In these cases, the nurse may neglect these neonates and deprive them of receiving quality care while they are still alive. However, there should be clear regulations to inform everyone about the formal and legal procedures to follow*

(nurse-18).

*Lack of a defined regional neonatal viability threshold.* According to the participants, there was no specific threshold determining the survival chances for newborns in any specific location within the country that impacted the involvement of parents in DM. One participant stated: *"A neonate born at 26 weeks of gestational age in a hospital in the capital may receive adequate care and be discharged, but what about in a remote area lacking facilities? In such a setting, the neonate may receive care and survive but could experience serious long-term sequelae. Regulations on viability thresholds should be tailored to the available facilities in each region and parents should decide about it"* (nurse-18).

*Lack of maternal guardianship of child medical care.* The lack of maternal guardianship of child medical care was another factor that impacted parental involvement in DM. According to participants fathers, as legal guardians of infants, had the authority to make decisions regarding their neonates' treatment, while mothers did not possess this official and legal right. Informed consent was obtained in the presence of fathers and with their signatures, and this legal limitation hindered mothers' participation in DM. One participant stated: *"It's unfair that the mother can't sign the consent form. I don't agree with this. The mother also has the right to make decisions, as she carries the child for 9 months with difficulty. Why doesn't the mother have the right to decide on the treatment, It doesn't seem fair that only the father is required to sign the consent form for treatment. I strongly oppose this"* (neonatologist-14).

**Deficiencies of the health insurance system.** Issues with insurance services were also identified as one of the factors that came from the 2 categories.

*Covering the cost of ineffective treatments.* Extensive health insurance coverage for ineffective treatments made it easier for doctors and parents to continue using these treatments, impacting their DM process. In this context, participants held the following beliefs. One

participant stated: *"Insurance covers the expenses, relieving parents of concerns about the costs of ineffective treatments. They prefer to try everything for their child and see the outcome. This affects their decision-making, as they may choose to bring the child home and request to stop ineffective treatments sooner if they have to spend money. Ultimately, insurance coverage frequently has a contrary impact in this situation"* (nurse-15).

*Lack of insurance covering palliative care services.* As per the individuals involved, the absence of regulation for insurance coverage of palliative care was also identified as a flaw that impacted parents' involvement in DM. The participants stated: "*When insurance covers the cost of medicine and equipment, more costly drugs are typically utilized. However, if parents have to pay, the situation completely shifts*" (Insurance company agent-19).

"*There is an immediate necessity for a palliative care team; however, the challenge is the lack of insurance coverage for the associated expenses. We have situations where a mother suffering from depression requires psychological therapy, but insurance does not include coverage for this type of counseling*"

(neonatologist-17).

**Treatment-oriented culture in society.** The study findings indicated that the treatment-oriented culture of society was also a factor that influenced how parents participated in making decisions for neonates with life-threatening conditions. This subcategory came from 3 categories.

*Expecting a miracle for medical science.* According to the participants, the excessive expectations of society and the overwhelming pressure to save lives at any cost hindered parents from being able to promptly participate in DM. One participant stated: *"The doctors may be forced to continue costly and ineffective treatments. It's common in society, people usually anticipate the doctor to prescribe pills or injections for them, even if they don't need it. However, this perspective needs to be changed. Perhaps numerous families with neonates suffering from incurable illnesses are seeking ways to alleviate their child's pain"* (neonatologist-23).

*Difficult acceptance of neonatal death.* Participants found it challenging to accept the death of newborns in society, which was another aspect of the culture focused on treatment that influenced parents' involvement in DM. One participant stated: *"People can't see a newborn baby die soon and expect it to live for many years"* (nurse-12).

*Difficult acceptance of home death.* The results indicated that the society's inclination to prolong medical treatment until the patient died in a hospital, and the reluctance to acknowledge death occurring at home, led to an emphasis on treatment and ongoing hospital care. This influenced the involvement of parents in DM. Participants stated: "*No matter what happens if it's God's will to take my baby, let it happen while we're still in the hospital. How can I bring her home?*" (mother-7).

"*I would rather the baby pass away in the hospital. Our home, where we have experienced two years of joy and laughter since getting married, would be harmful to us if the baby were to die there*"

(father-6).

**Physician-oriented culture in society.** The confidence that physicians possess in their DM authority, supported by a physician-centric societal culture, has also impeded parental involvement in DM processes. This subcategory was derived from 2 distinct categories..

*Excessive respect for physicians' decision-making eligibility.* Participants expressed the view that societal confidence in the DM abilities of physicians resulted in parents perceiving these professionals as more reliable than themselves in the context of DM, which in turn led to a reluctance to voice their own perspectives. One participant stated: *"It is more about people's perspective about patients and the overall societal attitude towards doctors. People believe that doctors have greater knowledge"* (neonatologist-22).

*Social position of physicians.* The esteemed professional and scientific standing of physicians conferred upon them enhanced societal influence and authority, thereby affecting parental DM and bestowing physicians with significant power. Participants stated: *"I don't usually consult the doctor because I say, well, she is a doctor, why should I disturb her time? I feel that she is busy and I don't want to bother her"* (mother-2).

*"In contemporary society, although nurses are equipped with considerable knowledge, physicians continue to maintain a position of authority. Parents often exhibit a higher level of trust in the physicians' sayings compared to those of nurses, even though both groups possess considerable knowledge and expertise in their respective fields.."*

(nurse-12).

## Discussion

This study explored the contextual factors that extended beyond the healthcare system and impacted parents' participation in DM for neonates with LTC. These factors encompassed legal and cultural aspects of society, which were classified into four subcategories and their categories.

The lack of regulations to modify ineffective treatments impacted parents' participation in DM. When a child needs extensive life-saving treatments despite a weak prognosis, there are often concerns about whether the interventions are futile. Medical futility occurs when the treatment is unable to provide its intended effect [21]. When the harmful effects of a treatment are greater than the possible advantages for a person, that treatment may no longer be in their best interest, even if it is not futile [22]. Infants are not able to communicate their likes and beliefs, so it is generally expected that parents will make decisions on their behalf in their best interest [23] and their values should guide the consideration of the potential harms and benefits [22]. A collaborative DM process is necessary, involving parents and clinicians working together to incorporate medical information and parental values [24]. This strategy receives support from policymakers and regulatory bodies globally [25]. The legal considerations for neonatal care may differ based on the neonate's location.

In the United States, most states have enacted legislation indicating that physicians are not required to provide treatment that is considered futile or lacking in medical efficacy [26]. Legal matters concerning the assessment of whether the disadvantages of an action are greater than the advantages are frequently debated using case law. It is strongly recommended to first attempt to resolve any potential conflicts within a team or between team members and families before resorting to legal requirements in all situations. In Iran as a Muslim country, Islamic ethics and Islamic law are seen as interconnected and cannot be separated [27]. In Islam deliberately ending a patient's life is viewed unlawful and as a form of homicide [28]. Nayeri et al. (2019) indicated that in Iran, the absence of explicit regulations in national legislation and the medical code of ethics concerning the initiation, continuation, or cessation of life-saving treatments has led many practitioners to encounter a legal void, prompting them to make decisions

influenced by their religious or moral convictions [29]. Furthermore, according to Banazadeh et al (2020), the risk aversion exhibited by healthcare professionals, along with their conservative practices, significantly impacted the involvement of parents in decision-making. Such conservative behaviors stemmed from concerns about potential litigation and fear of being accountable to the parents [13]. In Iran, while the survival rate of neonates facing life-threatening conditions has improved, initiating timely and clear communication with parents regarding the ineffectiveness of the treatment is challenging. Parents are often unprepared to transition from aggressive treatment methods to more compassionate, non-invasive alternatives. The unavailability of formal palliative care options and the absence of a palliative care ward or nursery for newborns caused continued hospitalization and continued ineffective treatment of the newborn [12]. Additionally, the education on palliative care, including neonatal and pediatric palliative care, is not incorporated into the clinical training or academic curriculum for nursing in Iran. The Bachelor of Science in Nursing (BSc) program includes only 2 to 4 hours of theoretical instruction on death and the care of deceased individuals [30]. Recently, a single credit unit titled "Ethical, Legal and Professional Principles in Newborn Care" has been incorporated into the curriculum for the Master's degree in neonatal intensive care nursing. In a cross-sectional study conducted in Iran in 2022, hospital records of twenty infants with severe hereditary metabolic disease were reviewed. The study revealed that although in 8 cases (40%), the parents (fathers) did not consent to the continuation of their child's treatment, in 7 cases (87.5%), the physicians continued the treatment (frequent sampling, medication, blood transfusion, and counseling for central vein placement) unchanged. Only in one neonate, blood sampling was stopped and the child was provided pain relief and comfort. In all instances where parental consent for the treatment measures was not obtained, there was no convening of a multi-specialty consultation meeting or a hospital committee to resolve the conflict between the physicians and the parents, nor to reach a consensus regarding the decision. Despite the parent's consent to discontinue treatment, some procedures were performed that included the collection of several blood samples and the administration of medications, blood transfusions, and consultations concerning the placement of a venous cutdown catheter. The authors stated that the absence of specific instructions for such conditions results in the treatment team continuing with ineffective treatments [31].

The lack of a legally documented DNR order also impacted parents' participation in DM. Several studies in Iran have indicated a prevalent disapproval of the DNR order [32, 33]. However, informal and verbal DNR orders in clinical settings are issued, which resulted in legal, ethical, and operational difficulties [33]. According to a study by Cheraghi et al. in 2016, which focused on the experiences of Iranian physicians with DNR orders, all individuals involved in the research viewed the absence of legal assistance as the primary reason for not adhering to the DNR directive. They suggested that creating a clinical guideline supported by legal measures could address numerous issues related to the DNR order to help prevent the provision of ineffective and unnecessary care [34].

In this study, participants believed the lack of a defined neonatal regional viability threshold impacted parents' participation in DM. Periviability, also known as the limit of viability, is the earliest point of fetal development (typically occurring between twenty-two and twenty-four weeks of pregnancy) when there is a chance, though not necessarily a strong probability, of the newborn surviving [35]. According to the Iranian National Maternal and Neonatal Registration Network (IMAN), At twenty-three weeks of gestational age, only 5.7% of infants survived, and the mortality rate was approximately 50% at twenty-eight weeks of gestational age [36].

There is no consistent standard for determining the viability of all fetuses in all settings and situations, especially one that relies solely on gestational age [37]. When there's a significant risk of death or severe morbidity, parents should be informed about the realistic options and

their associated risks and benefits. It's crucial to seek the parents' perspectives on resuscitation and consider the best interests of the newborn when making decisions [38]. Decision-making for a newborn at the edge of viability should be informed by the most reliable evidence regarding the newborn's prognosis. It's important to consider all relevant factors and not solely rely on gestational age [39]. In all decisions, the level of regional perinatal care services should be considered. In a research conducted in an eastern city of Iran, researchers discovered that newborns at a gestational age of twenty-seven weeks and three days were found to be capable of survival. They emphasized that this finding is specific to that region and may not apply to other areas within the country [40]. Akrami and colleagues (2021) argue that there is a need for enhanced care to improve outcomes for premature newborns in Iran. While the study participants noted that the absence of a clear threshold for neonatal regional viability is a barrier to parental involvement in DM, implementing a perinatal regionalization program could improve resource utilization and outcomes for newborns.

Furthermore, the lack of maternal guardianship of child medical care impacted parental involvement in DM. Guardianship, known as wilayah, is a legal status that allows an authorized individual to manage the affairs of someone unable to do so themselves. In essence, this person, known as a wali, has the power to act on behalf of another. In Shia law, the father and paternal grandfather are granted the authority of guardianship over the child [41]. According to Fairhurst and Long 2020 in a review study of parental involvement in DM in NICU, some fathers believed that mothers should have greater influence in DM, particularly regarding the newborn's care, as it would impact them more. It was also discovered that mothers typically made the final decision [42].

Furthermore, shortcomings in the healthcare insurance system affected parental involvement in DM. Participants felt that the insurance coverage for the expenses of diagnostic and therapeutic measures led to the newborn being hospitalized and receiving prolonged ineffective treatments, which made it difficult to consider transitioning to palliative care. However, the lack of coverage for home care and palliative care expenses hindered the discharge of the newborn and utilize home care and hospice services. Delivering home-based palliative care leads to better utilization of resources and cost savings [43]. This approach decreases expenses by transitioning from costly hospital care to coordinated and less expensive outpatient care [44]. While in many countries, insurance companies cover the costs of palliative care services, in Iran, insurance companies do not pay for hospice and palliative care [45].

According to participants, treatment-oriented culture in society affected how parents were involved in DM. They faced pressure from the public expectation to rely on medical science for miraculous treatments, which impacted their viewpoints and practices. Decision-making is influenced by the cultural context [46]. Based on Mirlashari et al., 2020 the social and cultural environment influences how parents perceive and expect to be involved in caregiving, as well as shaping the attitudes of healthcare professionals towards parental involvement [47]. Different cultures have a significant impact on the ethical decision-making process in crucial medical situations. The perception of what is ethical or unethical in healthcare can vary depending on the socio-cultural environment [14]. Parents' decisions about newborn care and their preferences for continuing or discontinuing treatment are shaped by their cultural background, values, religious beliefs, public awareness of the healthcare system, and societal norms [48]. Excessive optimism among parents about the potential for treatment is fueled by unrealistic hopes for technological advancements in society [49]. Iranmanesh et al, 2014 found families in Iran are unwilling to accept a poor prognosis for their child [30].

Additionally, the challenging acceptance of neonatal death and neonatal deaths occurring at home was another cultural treatment-oriented factor that affected parents' participation in DM. The death of a child always brings about pain, sorrow, and hardship for both families and

communities [50] and is affected by the media, and public expectations to save sick and premature infants. Family DM and the prompt transition to palliative care are often influenced more by public perception and cultural norms. [51] On the other hand, according to Banaza-deh and Rafii 2020, there has been less focus on providing home-based and hospice care for neonates with LTC compared to hospital-based care [52]. As a result, there is limited literature on home deaths. It is necessary to develop strategies for organizing home-based neonatal palli-ative care and evaluating the advantages and disadvantages of home-based care versus hospi-tal-based care, as well as determining the preferred location for end-of-life care.

Physician-oriented culture in society also was another beyond the healthcare system that affected parents' participation in DM. Participants believed excessive respect for physicians' decision-making eligibility and the social position of physicians stopped parents from being involved in DM. Another factor outside of the healthcare system that influenced parents' involvement in decision-making was the physician-oriented culture in society. Participants felt that the high level of respect for physicians' authority and social status hindered parents from taking an active role in DM. Some research studies support our findings. Many Iranian patients are not aware of their right to be informed and participate in decisions regarding their health [53]. Individuals frequently adopt a passive stance and perceive themselves as incapable of questioning the decisions made by physicians [54]. Many do not feel confident enough to make decisions for their children [53]. Mirlashari et al 2020 in a study on the challenges of implementing family-centered care in the NICU highlighted the significant influence of the unquestioned authority of physicians that parents tend to defer DM to them, indicating a lack of confidence in their ability to make decisions about their critically ill newborn. Participants recognized the historical context of physicians' dominance in the NICU, citing their powerful position in Iranian society as a contributing factor [47].

## Conclusion

We explored the external factors that affected parents' participation in DM for neonates with LTC conditions, beyond the confines of the healthcare system. Our findings revealed that the understanding of concepts like futility, DNR orders, the threshold of viability, maternal guard-ianship, parental autonomy, neonatal death acceptance, and medical technology varied depending on social, legal, cultural, and financial factors. Addressing the discrepancies between healthcare professionals' attitudes and religious beliefs, fatwas, and laws requires a collaborative approach. To navigate the complex ethical dilemmas surrounding neonates with LTC conditions and update laws to align with advancements in neonatology, input from a range of stakeholders such as clinicians, legal experts, Islamic scholars, sociologists, jurists, judges, and medical ethicists is essential.

## Limitations and recommendation

The variety of participants in the study was a positive aspect, but potential bias from research-ers is a common drawback in qualitative research. To address this, the researchers employed techniques such as member checking and collaborated closely with co-researchers.

## Acknowledgments

The authors express their gratitude to all participants who contributed their valuable experi-ences, which greatly assisted in this endeavor.

## Author Contributions

**Conceptualization:** Fatemeh Oskouie, Marjan Banazadeh.

**Data curation:** Fatemeh Oskouie, Marjan Banazadeh.

**Formal analysis:** Fatemeh Oskouie, Marjan Banazadeh.

**Methodology:** Fatemeh Oskouie, Sedigheh Khanjari, Marjan Banazadeh.

**Project administration:** Marjan Banazadeh.

**Supervision:** Marjan Banazadeh.

**Validation:** Marjan Banazadeh.

**Writing – original draft:** Marjan Banazadeh.

**Writing – review & editing:** Fatemeh Oskouie, Sedigheh Khanjari.

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
