## [Decision Letter · Decision Letter 0]

7 Aug 2024

PONE-D-24-09858Beyond the Healthcare system: The societal and contextual factors impacting parents’ participation in decision-making for neonates with life-threatening conditions Parent’s participation in healthcare decision-makingPLOS ONE

Dear Dr. Banazadeh,

Thank you for submitting your manuscript to PLOS ONE. After careful consideration, we feel that it has merit but does not fully meet PLOS ONE’s publication criteria as it currently stands. Therefore, we invite you to submit a revised version of the manuscript that addresses the points raised during the review process. The manuscript was well evaluated but there is an usefull suggestion I think the authors must observe. Please respond and include the suggestion and the manuscript may be accepted.

We look forward to receiving your revised manuscript.

Kind regards,

Ricardo Q. Gurgel, PhD

Academic Editor

PLOS ONE

2. We note that your Data Availability Statement is currently as follows: [All relevant data are within the manuscript]

Reviewers' comments:

Reviewer's Responses to Questions

**Comments to the Author**

1. Is the manuscript technically sound, and do the data support the conclusions?

Reviewer #1: Yes

Reviewer #2: Yes

2. Has the statistical analysis been performed appropriately and rigorously? 

Reviewer #1: Yes

Reviewer #2: Yes

3. Have the authors made all data underlying the findings in their manuscript fully available?

Reviewer #1: Yes

Reviewer #2: Yes

4. Is the manuscript presented in an intelligible fashion and written in standard English?

Reviewer #1: Yes

Reviewer #2: Yes

5. Review Comments to the Author

Reviewer #1: A brief description of the local health system structure is necessary since in the results it appears as one of the analysis categories and the American System is mentioned in the discussion to compare.

Reviewer #2: The study provides a good contribution regarding factors that impact parental participation in medical decisions related to critically ill newborns and this aspect is best addressed with qualitative research. Despite the small sample, all steps of the method are adequately described, with a correct explanation of possible biases. The results adequately support the conclusions and achieve the proposed objectives.

6. PLOS authors have the option to publish the peer review history of their article (what does this mean?). If published, this will include your full peer review and any attached files.

Reviewer #1: No

Reviewer #2: **Yes: **Aline de Siqueira Alves Lopes

---

## [Author Response · Author response to Decision Letter 0]

13 Aug 2024

Reply to comments ID PONE-D-24-09858

Translation and psychometric evaluation of the Persian version of the Nurse Clinical Reasoning Scale: A methodological study 

First of all, we would like to express our gratitude to the editor and the reviewers for their constructive comments and helpful suggestions. We greatly appreciate the careful reading of our manuscript. We have considered your comments and corrected the manuscript accordingly. A point-by-point response to the referees’ comments is given below:

If you have any questions about this paper, please do not hesitate to let me know. My e-mail address is:

banazadehmarjan@gmail.com

Comments Ref/ Page

 Reviewer #1: A brief description of the local health system structure is necessary since in the results it appears as one of the analysis categories and the American System is mentioned in the discussion to compare.

I would like to express my gratitude to the esteemed reviewer for their insightful comments.

The recommended information was added and the highlighted details can be found on Pages 17, lines 13-23, 18, lines 13-23,19, and lines 1-6.

 Reviewer #2: The study provides a good contribution regarding factors that impact parental participation in medical decisions related to critically ill newborns and this aspect is best addressed with qualitative research. Despite the small sample, all steps of the method are adequately described, with a correct explanation of possible biases. The results adequately support the conclusions and achieve the proposed objectives.

 I would like to express my gratitude to the esteemed reviewer for their insightful comments.

---

## [Editor Report · Decision Letter 1]

21 Aug 2024

Beyond the Healthcare system: The societal and contextual factors impacting parents’ participation in decision-making for neonates with life-threatening conditions Parent’s participation in healthcare decision-making

PONE-D-24-09858R1

Dear Dr. Banazadeh,

We’re pleased to inform you that your manuscript has been judged scientifically suitable for publication and will be formally accepted for publication once it meets all outstanding technical requirements.

Kind regards,

Ricardo Q. Gurgel, PhD

Academic Editor

PLOS ONE

Additional Editor Comments (optional):

You have responded correctly the suggestions and the manuscript is ready for publication.
---

## [Editor Report · Acceptance letter]

28 Aug 2024

PONE-D-24-09858R1 

PLOS ONE

Dear Dr. Banazadeh, 

I'm pleased to inform you that your manuscript has been deemed suitable for publication in PLOS ONE. Congratulations! Your manuscript is now being handed over to our production team.

Kind regards, 

on behalf of

Professor Ricardo Q. Gurgel 

Academic Editor

PLOS ONE